# Potentials of *Musa* Species Fruits against Oxidative Stress-Induced and Diet-Linked Chronic Diseases: In Vitro and In Vivo Implications of Micronutritional Factors and Dietary Secondary Metabolite Compounds

**DOI:** 10.3390/molecules25215036

**Published:** 2020-10-30

**Authors:** Barnabas Oluwatomide Oyeyinka, Anthony Jide Afolayan

**Affiliations:** Medicinal Plants and Economic Development (MPED) Research Centre, Department of Botany, University of Fort Hare, Alice 5700, South Africa; barnabastom@yahoo.com

**Keywords:** biomechanism, chronic diseases, dietary compounds, medicinal plants, *musa*, micronutrients, secondary metabolites

## Abstract

Nutritional quality and the well-being of the body system are directly linked aspects of human survival. From the unborn foetus to adulthood, the need for sustainable access to micronutrient-rich foods is pertinent and the global consumption of banana and plantain fruits, in effect, contributes to the alleviation of the scourge of malnutrition. This review is particularly aimed at evaluating the pharmacological dimensions through the biological mechanisms of *Musa* fruits in the body, which represent correlations with their constituent micronutrient factors and dietary polyphenolic constituents such as minerals, vitamin members, anthocyanins, lutein, α-,β- carotenes, neoxanthins and cryptoxanthins, epi- and gallo catechins, catecholamines, 3-carboxycoumarin, β-sitosterol, monoterpenoids, with series of analytical approaches for the various identified compounds being highlighted therein. Derivative value-products from the compartments (flesh and peel) of *Musa* fruits are equally highlighted, bringing forth the biomedicinal and nutritional relevance, including the potentials of *Musa* species in dietary diversification approaches.

## 1. Introduction

Micronutrient deficiency has developed into a key issue in developing and third world nations, with more than two billion people affected globally and expectant mothers and children being the majorly affected group [1,2]. Infants require close monitoring because they are in the sensitive growth and development phase of life [3]. Micronutrient deficiencies have negative health effects such as retardation of growth and development, compromised immunological system, weakened neuromotor efficiency and ultimately mortality [4]. Furthermore, micronutrient deficiencies result in lowered intelligence quotient levels, reduced work capacity and lower earning power [5,6]. Micronutrient deficiency is essentially a latent form of hunger which sets in due to disruption of absorption processes as a result of infection or disease conditions [7]. Poverty is a focal factor in low- and middle-income nations and it deepens the burden of micronutrient deficiency, just as poor dietary patterns (low intake of fruits) in the lower social class of developing countries are equally a liable factor [6,8]. Fruits offer chemopreventive potentials against malignant cancerous cells due to their biochemical constitution [9,10,11]. Population-based studies have indicated an inverse relationship between fruit consumption with cancer mortality and its recurrence [11,12,13]. Dietary administration of fruits has also been reported to be effective in the inhibition and chemoprevention of tumor malignancy [14]. Several studies have also identified the chemopreventive potential of fruit juices against cancerous cell lines [14,15,16]. An inverse association has been identified with fruit micronutrients and deposition of fat and cardiovascular disease among obese populations [17,18,19]. Similarly, epidemiological studies widely indicate a dose-related link between fruit intake and chronic diseases [20,21,22,23,24]. In relation to the forgoing discourse, a number of fresh fruits were surveyed and obtained from supermarkets, and are shown in Figure 1.

## 2. Overview of the *Musa* genus

The genus *Musa* is found in the Musaceae family. Other members of the family are the *Ensete* and *Musella* [25]. They are essentially monocots. Carl Linnaeus (1707–1778), the taxonomist, grouped bananas by their dietary utility: *Musa paradisiaca* (plantain); *Musa sinensis* (banana). A comprehensive taxonomic classification of the genus *Musa* is shown in Figure 2.

The *Musa* plants range from two to nine metres in height (cultivated) and 10–15 m (wild), consisting of the false stem (pseudostem), corm, foliar including the flowering part.

Banana (*M. sinensis* L.) is a tree-like, perennial herb. The term banana was developed out of “banan” of Arab origin (i.e., “finger” in translation) [26]. It was introduced to parts of western Africa (Guinea) from Portugal, with the first domestication of banana in South-eastern Asia [27]. The plantain (*M. paradisiaca* L.), similar to the unripe banana, is larger, has a starchy flesh and can be used in the unripe and cooked forms. It is a tropical staple, ranked the tenth most important in the world and contains more starch and less sugar than a banana. The mature, yellow variety can be peeled as is typically done with banana.

The Banana family, broadly used to describe the *Musa* genus and its herbaceous members, including the fruits they bear, have flowers that are medicinally useful in the treatment of bronchitis, dysentery (gastro-intestinal infection bowel movement) and ulcers. Diabetic patients are traditionally served cooked flowers, while the astringent sap is usually applied for conditions such as hysteria, epileptic seizures, fever, leprosy and diarrhoea [28]. It is also useful in cases of haemorrhoids, bites and stings. Young leaves are applied as poultices on burns and other skin disorders, while the leaf and unripe peel ashes are taken for digestive conditions like diarrhoea and dysentery including the treatment of pernicious ulcers. In India, roots and seed mucilage are administered to ameliorate disorders of the digestive system [28]. Banana peel protects the fruit, contributes a minimum of 30% to the total weight of the banana and contains substantial quantities of phosphorous and nitrogen [29]. It also supplies key nutritional mineral elements like magnesium and potassium [30], and contains higher phytochemical constituent in the peel than the flesh (pulp) [31]. In sub-Saharan Africa, more than 30 million people are estimated to feed on bananas as their main energy source as most African breeders and growers still operate at the subsistent level. There is a rising investigative pattern in the use of unripe banana products by consumers for their polyphenol content [32], anthocyanin in the flesh [33] and antioxidative potential in unripe banana flour [34,35]. Studies from literature reveal that non-commercial cultivars have higher levels of antioxidative propensity [36,37]. Plantain is reliably a year-round staple, especially in third world nations laden with inadequate technologies in the frontiers of storing, preserving and transporting food products. On the African continent, plantain fruits meet about one-quarter of the carbohydrate requirements of 70 million people thereabout. It is a versatile culinary raw material for products like chips or dodo (baked or roasted), fufu, porridge, flour eaten with soup, or eaten alone depending on the consumer’s taste [38]. Banana and plantain fruits are quite rich in dietary fibres, which is essential for the optimal functioning of the gastro-intestinal and digestive system [30,39]. Studies have reported higher nutritional profiles in *Musa* fruit peels compared to a number of fruit peels such as *Mangifera indica*, *Carica papaya*, *Citrus sinensis*, *Ananas comosus*, *Malus domestica*, *Citrullus lanatus* and *Punica granatum* [40]. A detailed comparative profile of nutritional factors with regards to the aforementioned is indicated in Table 1.

## 3. Micronutrients

They broadly encompass trace elements, vitamins and mineral elements key to regular cellular and molecular functioning, which makes them of significance to human health in spite of their small requirement levels [41,42,43]. Micronutrient deficiency alleviation is vital to the prevention of chronic disease and mortality amongst deficient populations [44,45]. A number of micronutritional constituents have been identified in the compartments of *Musa* fruits as shown in Table 2.

## 4. Biomechanismal Implications of Micronutritional Factors of *Musa Species* Fruits

### 4.1. Antioxidant Mechanism

Zinc, structurally depicted in Figure 3, is involved in cellular proliferation, differentiation and apoptosis. It prevents the formation of free radicals and cushions the side effects of anti-inflammatory mechanisms [52,53,54]. Zinc deficiency increases inflammatory cytokine levels, oxidative stress as well as cellular dysfunction [55]. Zinc exhibits its antioxidant capacity through the induction and inhibition of hemeoxygenase and NADPH oxidase respectively [56]. Similarly, zinc has been reported to deplete biomarkers of oxidative stress on the basis of human mononuclear cell experiments [56]. It is also involved in the upward regulation of the gene expressions which encode for antioxidant and detoxification molecules [55]. The antioxidant effect of zinc has also been identified in the mechanism of protein sulfhydryl stabilization which is targeted at nullifying oxidation [55]. Zinc protects the cells of the endothelium from reactive oxygen species like hydrogen peroxide through the biological synthesis of glutathione which is stimulated by the Nrf2 (nuclear erythroid 2) factor [57].

### 4.2. Blood Sugar Balance and Immunity 

Dietary zinc is involved in the synthesis and secretion processes of insulin within the β-cells of the pancreas, on the basis of the β-cell viability role of zinc in an animal model [58], while it contributes to immunity by mediating in infection resistance and activating the T lymphocyte [59,60]. Animal experiments have confirmed the role of zinc in immunological memory on the basis of antibody recall reactions to T-dependent and independent antigens [61,62].

### 4.3. Neurological Activity

Dietary zinc acts as a neurological transmitter in the brain and is functionally key in cognition, learning and memory [63] and activates the neuron-based Erk signal mechanism [64]. The regulation of memory formation has been depicted by zinc via the Erk pathway [65].

### 4.4. Immunological, Neurological, Antioxidant and Anticancer Mechanism

Dietary boron, structurally shown in Figure 4 is biologically essential in a number of modes such as improved central nervous system and enhanced immunity. A number of in vivo and human studies have pointed out that dietary boron is functional in the activity of the brain with its deficiency being associated with a drop-in electrical activity [66]. Furthermore, dietary boron deprivation studies have indicated reduced high-frequency and a spiked low-frequency electric activity in the brain [66]. In the same vein, low boron intake (<0.3 mg/d) assessments in human models reflected lowered cognitive-motor functions. It has been reported from in vivo investigations that boron has the capacity to reverse oxidative stress and free radical production usually triggered by endotoxins [67]. Furthermore, the antioxidative stress capacity of boron has been identified via β-cell preservation in the pancreas, including a dose-dependent pattern (5–20 mg/kg) in tissue damage amelioration [67]. In a similar mode, boron mitigates oxidative stress by increasing the oxidant-neutralizing glutathionic reserves [68]. The induction of boron reverses the oxidative stress expressed in cellular carcinoma of the hepatic system [69]. It also enhances metabolism in the hepatic system [70] and functions collaboratively with calcium, magnesium and phosphorus in the regulation of parathyroid function [71]. Anticancer inhibition against proliferative cell death in prostate and breast cancer cell lines (LNCaP, DU-145) have been reported in boric acid [72,73,74]. Furthermore, the non-tumor prostate cell lines (RWPE-1, PWR-1E) were inhibited in a dose-dependent form (100–1000 µM) by boric acid, while the PC-3 cancer cell line was experimentally inhibited at higher levels than the observed blood levels [72]. There is also the antiproliferative mechanism of boric acid identified in a dose-dependent (500–1000 µM) depletion of MAPK proteins [73].

### 4.5. Antioxidant Mechanism

Iron, depicted structurally in Figure 5, is an essential building block for red blood cell haemoglobin which invariably contributes to oxygen distribution from the lungs [75]. Iron deficiency is a very common nutrient disorder that causes anaemia. Iron is important in preventing the susceptibility of the placental unit of foetus from oxidative stress [76]. There is a body of thought that projects the antioxidant mechanism of micronutrients as oxidative stress reducers which enables improved antenatal and postpartum conditions [77,78].

### 4.6. Blood Sugar Balance and Antioxidant Activity

Manganese deficiency is not quite pronounced in humans as opposed to animals [79]. However, a significant chunk is lost in refined foods and so fruit intake presents a rich source of dietary manganese [80]. Manganese enhances blood sugar balance as a metalloenzyme constituent implicated in the synthesis of metabolic processes for glucose and glutamine [80]. It also exhibits antioxidant capacity as a component of the superoxide dismutase (SOD) which combats free radicals [81,82] and is significant because of its focal mitochondrial location in the protection of DNA and genetic make-up. Manganese is thus an important biochemical entity and its molecular structure shown in Figure 6.

Folate (Vitamin B_9_) enables red blood cell production (erythropoiesis). It enhances immunity levels and biosynthesizes nucleic acids, nerve tissues (brain functionality) and proteins [83,84]. It helps in DNA damage risk prevention and immunological homeostasis [85,86,87]. In addition, folate has epileptogenic (anti-seizure) properties and enhances lymphocyte repair of oxidative damage [88,89,90], identified the role of incremental folic acid levels (1 ng/mL–2 µg/mL) in stimulating in vitro human venous lymphocyte growth and reduced DNA strand breakage. Figure 7 shows the molecular structure of folate members.

It is a micronutrient essential for neurological and haematological functioning [91]. It is important in several metabolic processes because of its presence in a wide spectrum of enzymes. Dietary copper enables the synthesis of haemoglobin, neurotransmission, iron oxidation as well as peptide amidation-linked antioxidative defence [92]. Copper can scavenge or mop up free radicals by neutralizing them and in turn prevent their potential damage [93,94,95]. Dietary copper is also key to optimal skeletal functioning by enhancing collagen formation which ensures bone competence and rigidity [96]. In addition, copper functions in myelin formation which insulates nerve cells and triggers nerve impulse transmission [96]. Figure 8 indicates the molecular structure of copper.

Pantothenic acid, whose molecular structure is shown in Figure 9 is also known as Vitamin B_5._ This micronutritional factor is essential for the metabolism of fatty acid. Pantothenic acid has anti-stressor properties which also contribute to the production of neurotransmitters [97]. The body’s stress resistance capacity is boosted by pantothenic acid through the build-up of antibodies and it as well enhances central nervous system development [97,98].

### 4.7. Immunological and Anti-Inflammatory Mechanism

Riboflavin (Vitamin B_2_) is thought to be implicated in the differentiation and functionality of immune cells by regulating the oxidization of fatty acid [87]. Dietary riboflavin contributes to the generation of inflammatory and immunity signalling molecules within immune cells through the mechanism of NADPH oxidase 2 priming [99]. Riboflavin exhibits anti-inflammatory capacity by suppressing the nuclear factor (NF-kB) activity [100]. Its structural molecular entity is depicted in Figure 10.

### 4.8. Immunological, Antioxidant and Neurological Mechanism

Pyridoxine (Vitamin B_6_), structurally shown in Figure 11, enhances immunity integrity via the linkage formation between chemokines and cytokines, while it enhances immune feedback towards increased antibody output [101]. Dietary pyridoxine exhibits antioxidant activity via the inhibition of erythrocytic lipid peroxidation [102] and reduces the predispositionary risk to stroke and arteriosclerosis. It is involved in synthesizing haemoglobin and neurotransmitters as well as gluconeogenesis [97,103]. Pyridoxine controls the risk of acute coronary syndrome and athero-thrombosis by modulating blood homocysteine levels [104].

Thiamine (Vitamin B_1_) is centrally important in nerve functioning and energy generation from carbohydrates [97]. Thiamine modulates the neurological transmission system, improves brain functionality [105], protects the peripheral nervous system from compromise, and is involved in synthesizing myelin [106,107]. Its molecular structure is shown in Figure 12.

Niacin (Vitamin B_3_), shown in Figure 13, is anti-inflammatory and immunohomeostatic in its activity. It inhibits the multiplicity of pro-inflammatory cytokines and the tumor necrosis factor usually effected by monocytes and macrophages [108]. Niacin deficiency in the body is symptomized by the pellagra condition which disrupts the gastrointestinal and neurological system [109,110]. A number of studies have highlighted the biological potential of niacin in anaemic, hypertensive, cardiovascular, hepatic and cancerous disease conditions [111,112,113,114,115,116].

The presence of phytochemicals in plants has been identified for their potential pharmacological potentials [117]. Table 3 is a compendium of dietary secondary metabolites identified in fruit components of *Musa* species, which reflects the broad phytomedicinal benefits derivable from their dietary intake.

## 5. Biological Mechanism and Pharmacological Activity of Dietary Phytocompounds of *Musa* Species Fruits

Antioxidant micronutrients, like vitamins and carotenoids, are chief contributors to the defence mechanism against reactive oxygen species (ROS) in the body [145]. Other works have reported that antioxidant vitamin and carotenoid levels were low in hepatitis and cirrhotic liver conditions [146,147]. Carotenoids are accumulated majorly in the hepatic organ and are released into blood circulation as lipoproteins. They also participate in the antioxidative defence mechanism when present in the liver and in high concentrations of free radical species. As a consequence, carotenoid physiological functions could interact with or inhibit liver dysfunctions such as acute hepatitis, hepatic steatosis, chronic hepatitis, hepatic fibrosis cirrhosis, hepatocellular carcinoma [148]. Some carotenoid members (zeaxanthin, lutein, lycopene and astaxanthin) represented in Figure 14, help in preventing the development of non-alcoholic fatty disease of the hepatic organ by mechanisms such as improvement of insulin signalling, depletion of the influx of free fatty acids into the hepatic organ [149]. Furthermore, dietary carotenoids have the capacity to reduce aging-related diseases by deploying the mechanism of reactive oxygen species production in order to inhibit cellular dysfunction and oxidative stress. These are hallmarks of antioxidative capacity [150].

α-tocopherol has been implicated in non-antioxidant mechanisms such as modulation of cell functions [151], inhibition of platelet adhesion and aggregation for blood clotting and inhibition of cytokine release [152,153,154,155]. It also functions in phosphorylative regulation for prevention of cardiological conditions, due to its modification action on the proliferation of adhering cells and cellular oxidant production via vitamin E- specific pathways [156]. It also acts as an antioxidant in a chain-breaking mechanism against free radical propagation [156] and in the prevention of subfertility conditions (loss of spermatogenesis and poor zygote retention) on the basis on in vivo tests [157]. Tocopherols or tocotrienols are neuroprotective by inhibiting glutamate-induced death in neuronal cells [158]. Figure 15 shows the molecular structure of tocopherol members.

Catechins, shown in Figure 16, are anticancer phenolic compounds present in banana [159]. They function in mitochondrial cells in response to oxidative stress through enhanced phosphorylation, incremental production of ATP and preservation of mitochondrial membrane integrity [160]. Animal model experiments revealed corresponding effects of catechin on mitochondrial respiration and β-cell functioning [160]. Catechin has also been implicated in the upregulation of mitochondrial complexes which translates into increased ATP generation in the cell [161]. The anti-inflammatory mechanism of catechin is expressed through the modulation of transcription factors that are related to activated B cells (NF-κB) and activator protein-1 (AP-1). In vitro and animal model studies reveal dose-dependent apoptotic activity of catechin in the region of 50 µM [128,162] which further inhibits the overexpression of cyclooxygenase-2 (CoX-2) [129].

Coumarins have anti-inflammatory and antioxidant activity [163], they are anticoagulatory in the liver and are anticancerous by inhibiting the microtubule and arresting tumor cells [164]. They are also antiallergic and antiproliferative in their activity [165,166]. They exhibit strong pharmacokinetics on the basis of their easily absorbable and metabolizable nature [167]. In addition, coumarins readily function as antitumor agents and are significant because they have the capacity to counteract the side effects usually identified with the chemo and radiotherapeutic procedures [168,169,170]. Studies have identified the cytotoxic activity of coumarin compounds against leukemia cell lines (HL-60; NALM-6) and colon tumor-8 [171,172]. Figure 17 depicts the molecular structure of coumarin.

Phytosterols inhibit the absorption of cholesterol by depleting the collection of metabolizable cholesterol, thereby lowering the risk of cardiovascular disease [173,174,175,176,177,178,179]. β-sitosterol has anticancer activity as seen in in vivo evidence by inhibiting the development and proliferation of breast cancer cell lines [180,181]. Furthermore, it acts as a chemopreventive actor against colon, prostate, mammary and lung cancer cells by suppressing oxidative stress [182,183,184]. Some phytosterol members are structurally depicted in Figure 18.

Terpenoids are anti-inflammatory and immunomodulatory in action [185] with anticancer capacities projected through the mechanism of inhibiting tumor proliferation by augmenting the levels of tumor proteins, B cell lymphoma and deactivating Akt signalling [136,186,187,188]. Monoterpenoids have gastroprotective activity which has been identified through the modification of deleterious stress effects on gastro-intestinal injury, while limonene acts in the neutralization of stomach acids and enhancing proper peristalsis [189,190]. The molecular structure of terpenoids is depicted in Figure 19.

Anthocyanins are neuroprotective, antiobesity and antidiabetic [191,192]. They also inhibit the growth of malignant cancerous cells [193,194,195], with anti-inflammatory and cardiovascular protection capacities [196,197,198]. Furthermore, in vitro study shows that they suppress colorectal cancer cell lines (DLD-1 and COLO 205) by apoptosis [199]. Anthocyanin members are structurally shown in Figure 20.

Catecholamines trigger biological reactions and increased energy metabolism levels [144,200], which function as antistressors or stress repressants. The mechanisms of increased thermogenesis, energy expenditure and reduction of fat reservoirs that are triggered by catecholamines are thought to be mediated by the adrenergic β-(β3)-receptors [144]. Epinephrine and norepinephrine enhance glucose release into the bloodstream as a result of its glycogenolytic activity in the liver [201,202]. Figure 21 depicts the molecular structure of some catecholamine members.

## 6. Dietary Incorporation of Banana and Plantain (*Musa* spp.)

Can banana peels be eaten? This non-conventional issue has been addressed and the point is that indeed, banana peels can be eaten. Reference [203] noted that banana peels are consumed across various cultures and communities and they are simply too valuable to be perpetually disposed of. He further buttressed this point by asserting that indeed, banana peels can be eaten by human beings. Some people eat banana peels on the basis of the taste and textural modification it brings to the diet as well as the knowledge of the nutritional value present therein. Many fruits and food items may not be particularly pleasing to the eye and this, for some people is the case with banana peels as the yellow skin can be a put-off [203]. However, the many health benefits simply outweigh these sentiments. Banana peels, a major by-product of the banana processing industry are reportedly rich in fibre and nutrients, chief among are potassium, magnesium and calcium [204]. However, commercial utilization has not been maximised. Thus, the peels have notoriously become a dumping waste after consumption of the inner fruit pulp. This fate has also befallen the plantain peel, despite the huge potentials inherent. [51] has reported quite useful stores of cellulose (7.6% to 9.6%), hemicelluloses (6.4% to 9.4%) and lignin (6% to 12%) in the peels of banana. These also put banana peels in good stead as an aid in digestion processes. Creativity can be employed as the peels can be used to derive the therapeutic banana tea or perhaps a smoothie blend, just the same way the pulp has been applied as a banana smoothie and even the baked banana Madeira product (Figure 22). Essentially, the aim is to provide a more consumer-friendly outlook. Furthermore, some culinary measures have been put forward in a bid to improve the appeal of the peel products and they include: looking out for ripened banana fruits, the riper they are, the thinner and sweeter they tend to be, including thorough peel washing as a basic requirement [205]. Dieticians prescribe the intake of banana peels on the basis that the dietary fibre content ameliorates cholesterol levels in the bloodstream. Equally, plantain peels also have nutritive and dietary potentials as [206] have reported relevant protein, dietary fiber and antioxidant levels in plantain peel flour. The plantain peel flour has found utility as a substitute to wheat flour in the binding of sausage snacks [207]. [208] utilized the significant dietary fibre content in plantain peels in the derivation of cookies with high fibrous content. There is also the dietary option of incorporating plantain peel flour into pasta meal, traditionally identified with semolina flour [209]. There is also the plantain peel chutney, a local Indian recipe which is usually fried and prepared with condiments and taken in combination with a meal of rice (Figure 22). Essentially, dietary diversification potentials are inherent in banana and plantain fruits (Figure 22).

## 7. Production and Consumption Status of Bananas (*Musa* spp.)

Banana is an important subtropical and tropical fruit, which is generally cultivated in subsistence or small scale and economic or large scale across the globe as reported by [210]. According to [211], banana and plantain, members of the *Musa* genus are ranked fourth in export value after wheat, rice, and corn (4.5 to 5.0) billion United States Dollars (USD) per year from 1998 to 2000. According to the Food and Agricultural Organization (FAO), banana cultivation is evident in more than 130 nations and on more than 5.5 million hectares of land. A worldwide output running to 145 million tons has also been recorded [212]. In essence, with a 145-million metric ton production output worldwide (worth about £26.5 billion), *Musa* species are one of the globally most important staple food crops and arguably most popular international trade fruits, based on the Food and Agricultural Organization reports [213]. Reports from [214] assert that some two-thirds of the banana family cultivated and produced in west and central Africa are plantains, whose starch-rich fruits require culinary preparation for consumption, while dessert and cooking bananas make up the remaining one-third. In Africa, plantain is cultivated from Guinea to the Congo Democratic Republic and the Central African Republic. The countries with the majority production are the West African nations of Cameroon, Ghana, Nigeria and Ivory Coast (Table 4). Plantain production ranks highly in these regions (about 12327974 tonnes in 2014) among the starchy staples [214]. Plantain, as a reliable all-season staple, in developing nations faces issues of inadequate food storage, preservation and transportation technologies. In Africa, plantains and bananas supply over a quarter of carbohydrate requirements for more than 70 million people. Plantain averagely contains around 220 calories, useful potassium content and a source of dietary fiber. It is one of the major horticultural crops including the top ten most important crops in terms of food security globally and a consistent diet in rural areas and urban metropolises [215].

The global banana supply chain is a complex one, much dependent on several collaborations of parties. Most of the banana export stems from Ecuador, Guatemala and Costa Rica as of the year 2016. However, only about 15 to 20% of banana produced globally, end up traded in the global market. More than 1000 banana varieties are produced and consumed globally. However, Cavendish banana, accounting for 47% of global production recovers quickly from natural disaster shocks. An estimated 50 billion tonnes of Cavendish are produced yearly around the globe, and predominantly supplied to the United States of America and Europe, because they are indeed better suited to international (global) trade and possess more resilience to global travels. Precise production figures can prove a little difficult to pin down, especially because the cultivation of banana plants is majorly done by small scale farmers and trade in the informal sector. However, the available data point to the fact that between the years 2000 and 2017, the global production rate rose at about 3.2%, reaching 114 million tonnes by 2017 from 67 million tonnes in the year 2000, with Asia, the Americas and Africa being the chief producing regions (Figure 23). Bananas and plantains make up a major daily supply of carbohydrates for about 100 million Africans [216], exemplified by Uganda, with an average annual per capita consumption of 223 kg. The banana industry has made productivity improvements from 14 to 20 tonnes per hectare from 1993 to 2017 [217]. The major catalyst of increased production is the concomitant human population growth rate around the globe, surpassing seven billion people. This is, in particular, more clearly expressed in the increased consumption needs of developing countries.

**Table 4 molecules-25-05036-t004:** The Major Global Producers of Plantain (*Musa paradisiaca*).

Rank	Country	Production (Tonnes)
**1st**	Cameroon	4.31 million
**2nd**	Ghana	3.95 million
**3rd**	Uganda	3.71 million
**4th**	Colombia	3.54 million
**5th**	Nigeria	3.09 million
**6th**	Philippines	3.07 million
**7th**	Peru	2.07 million
**8th**	Ivory Coast	1.59 million
**9th**	Myanmar	1.11 million
**10th**	Democratic Republic of Congo	1.11 million

Source: [218].

## 8. Conclusions and Perspectives

This review identified and discussed a spectrum of micronutritional factors and dietary bioactive compounds in the fruit compartments (pulp and peel) of *Musa* species. These bioactive constituents inevitably confer a series of potential pharmacological values against oxidative damage, which helps in preventing genetic damage or DNA compromise, enhancement of immunological competence, neurological functioning and lowered risk of cardiovascular disorder. It is thus clear that banana and plantain fruit peels, as much as the flesh compartments, have a host of nutritionally and pharmacologically significant values to human nutrition, health and dietary quality. This consequently places banana and plantain peels in good stead as potential natural products and functional food options, adding credence to the need for their increased dietary utility. Summarily, given the activities described in this review, these primary and secondary metabolites exhibit properties that justify more research by the scientific community, to drive preclinical and clinical studies leading to the development of new drugs. The pharmacological insights in this review indicate the potential of *Musa* fruits as key elements in driving further preclinical research, in a bid to achieve natural product-based drug development from the fruits of *Musa* genus.

## Figures and Tables

**Figure 1 molecules-25-05036-f001:**
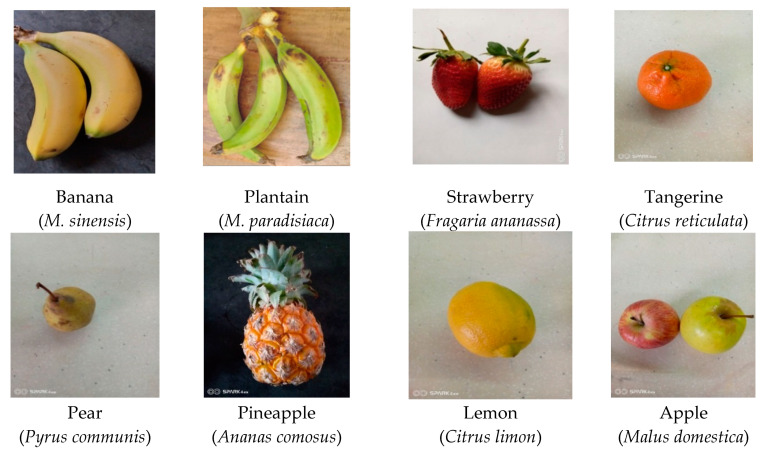
Photos of commonly consumed fruits including *M. sinensis* and *M. paradisiaca*.

**Figure 2 molecules-25-05036-f002:**
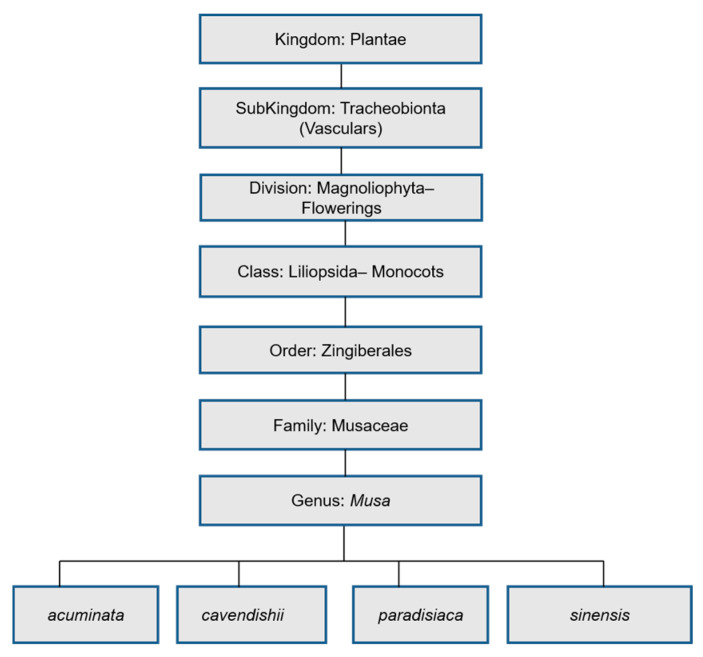
Scientific classification of *Musa* genus.

**Figure 3 molecules-25-05036-f003:**
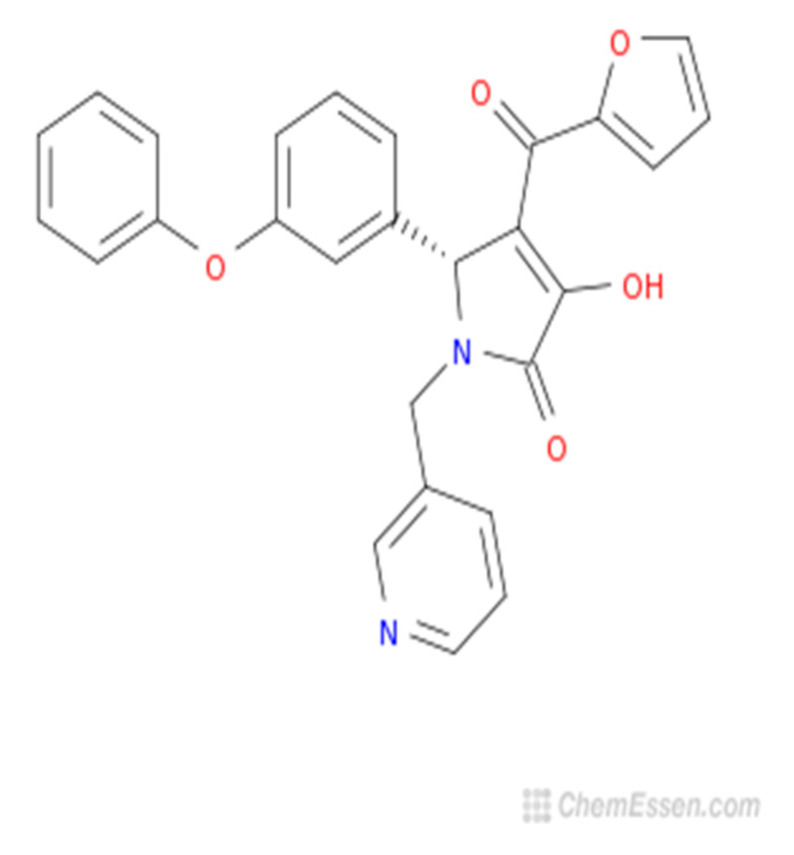
Zinc structure.

**Figure 4 molecules-25-05036-f004:**
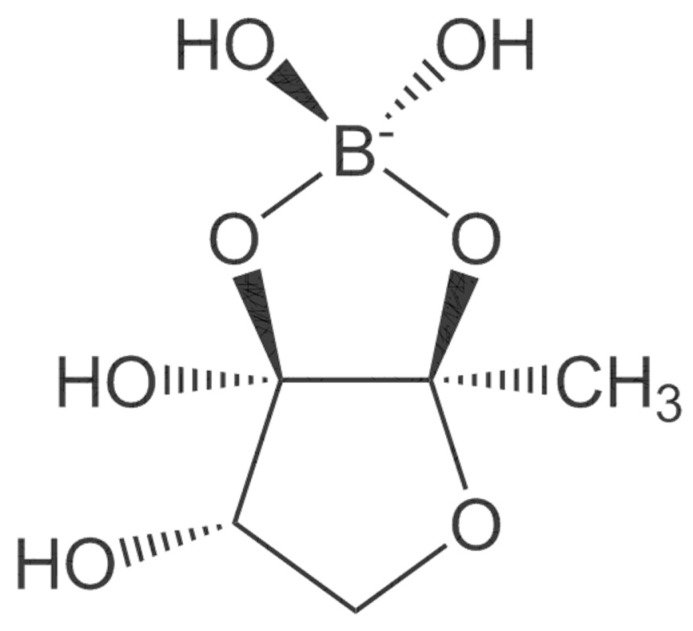
Boron structure.

**Figure 5 molecules-25-05036-f005:**
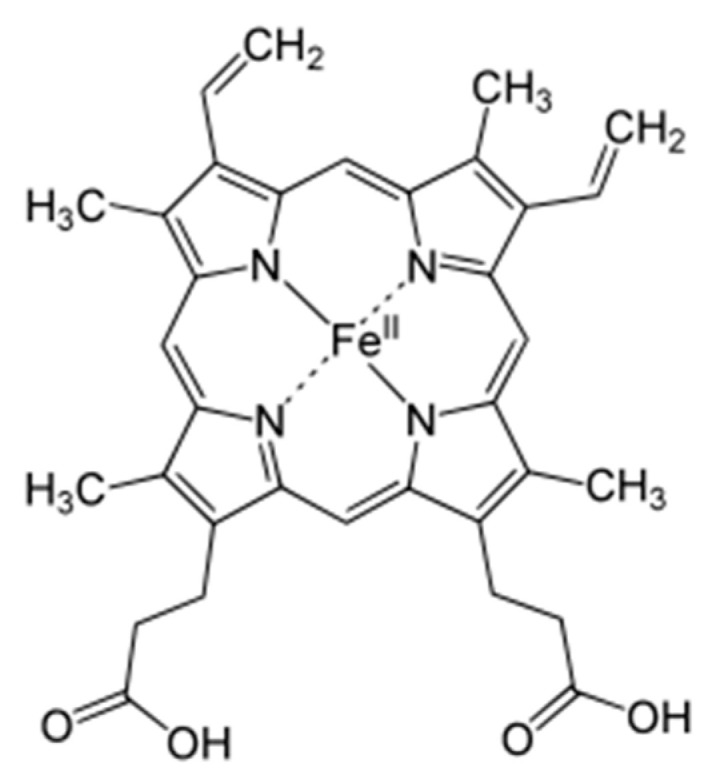
Iron structure.

**Figure 6 molecules-25-05036-f006:**
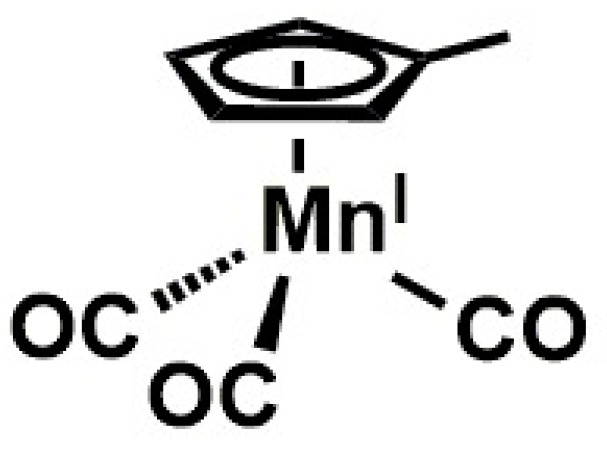
Manganese structure.

**Figure 7 molecules-25-05036-f007:**
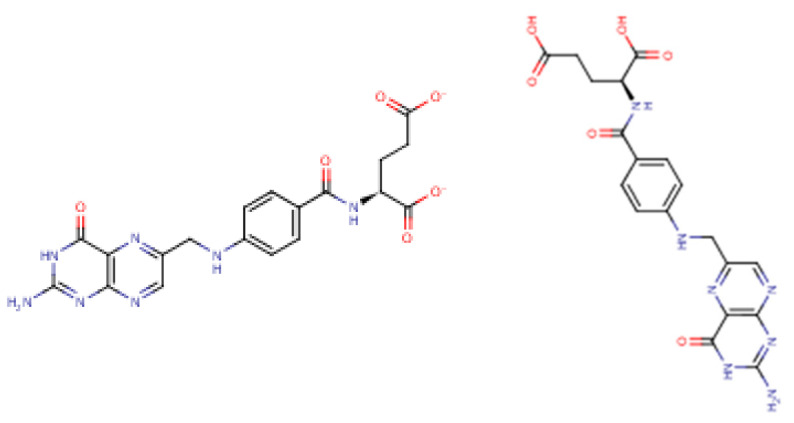
Structures of Folate and folic acid.

**Figure 8 molecules-25-05036-f008:**
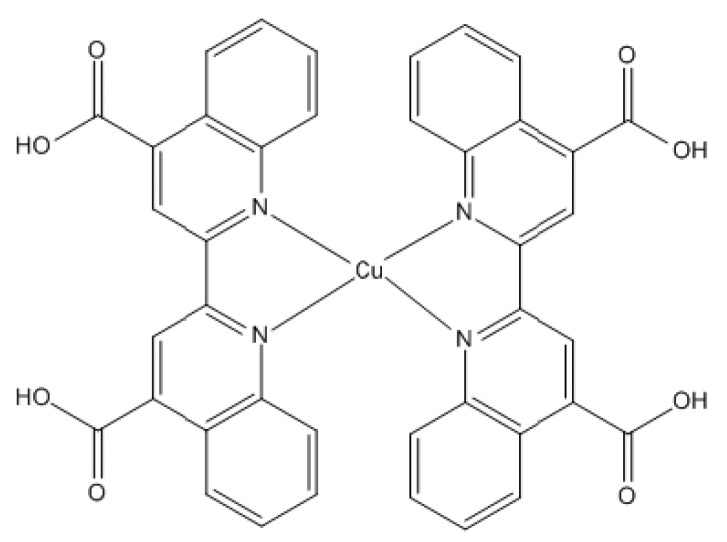
Copper structure.

**Figure 9 molecules-25-05036-f009:**
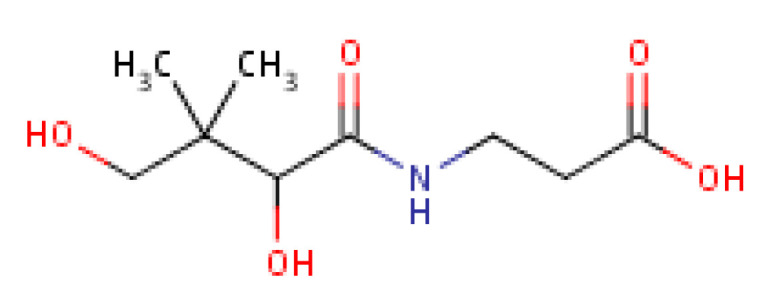
Pantothenic acid structure.

**Figure 10 molecules-25-05036-f010:**
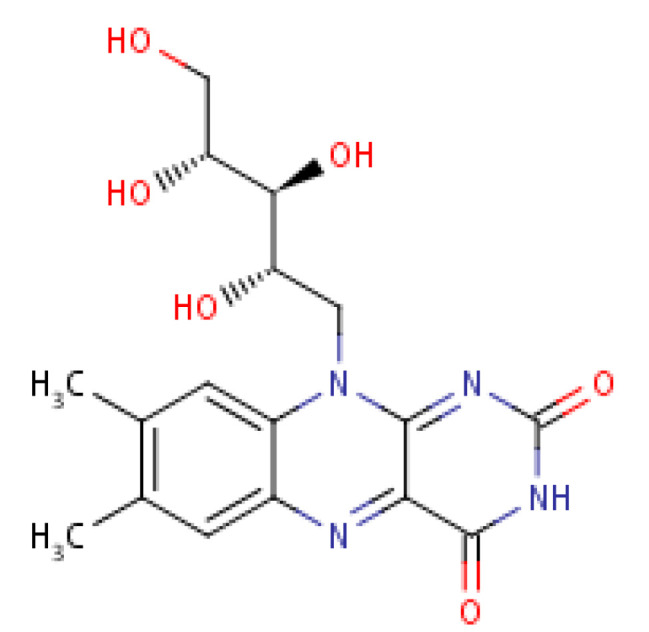
Riboflavin structure.

**Figure 11 molecules-25-05036-f011:**
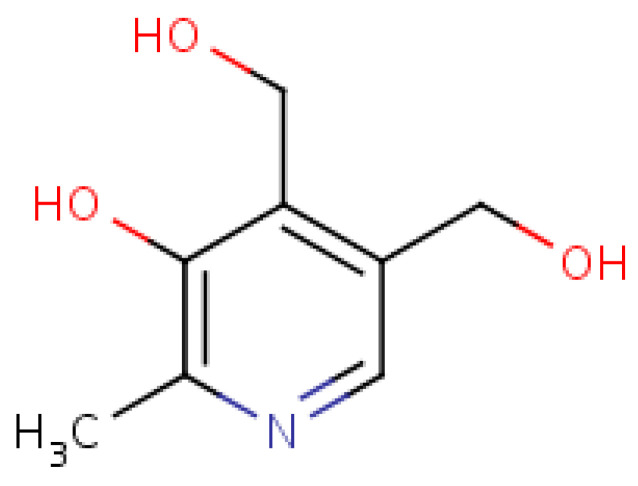
Pyridoxine structure.

**Figure 12 molecules-25-05036-f012:**
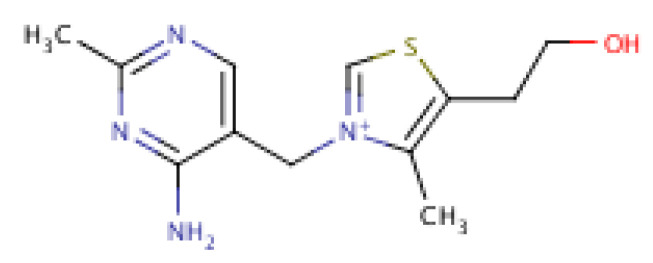
Thiamine structure.

**Figure 13 molecules-25-05036-f013:**
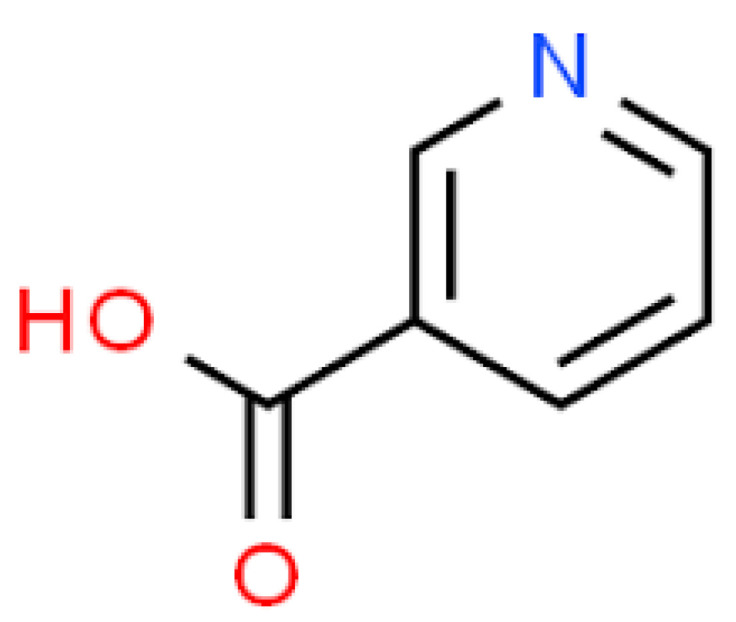
Niacin structure.

**Figure 14 molecules-25-05036-f014:**
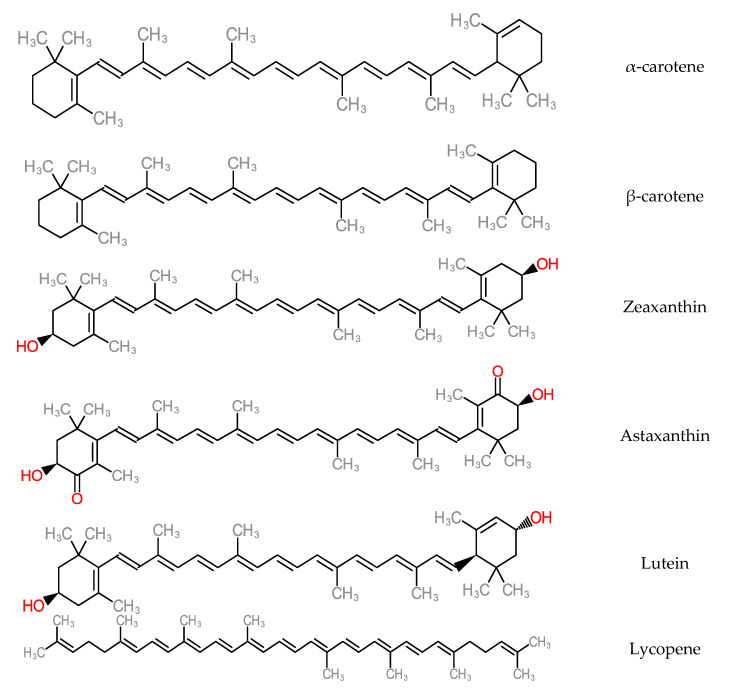
Structure of Carotenoids.

**Figure 15 molecules-25-05036-f015:**
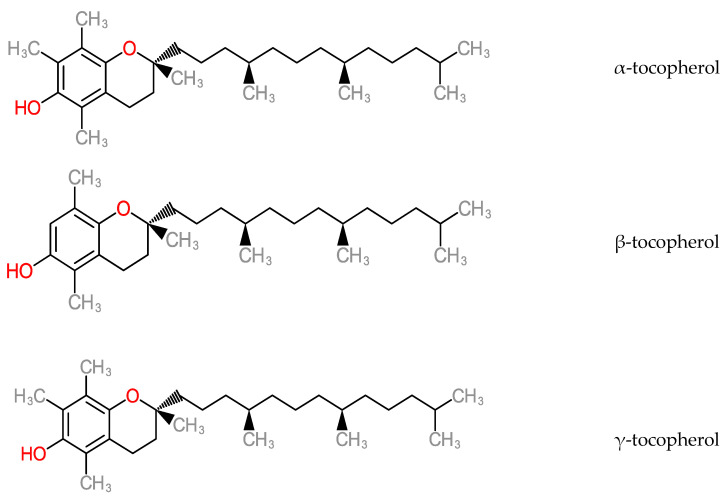
Structure of Tocopherols.

**Figure 16 molecules-25-05036-f016:**
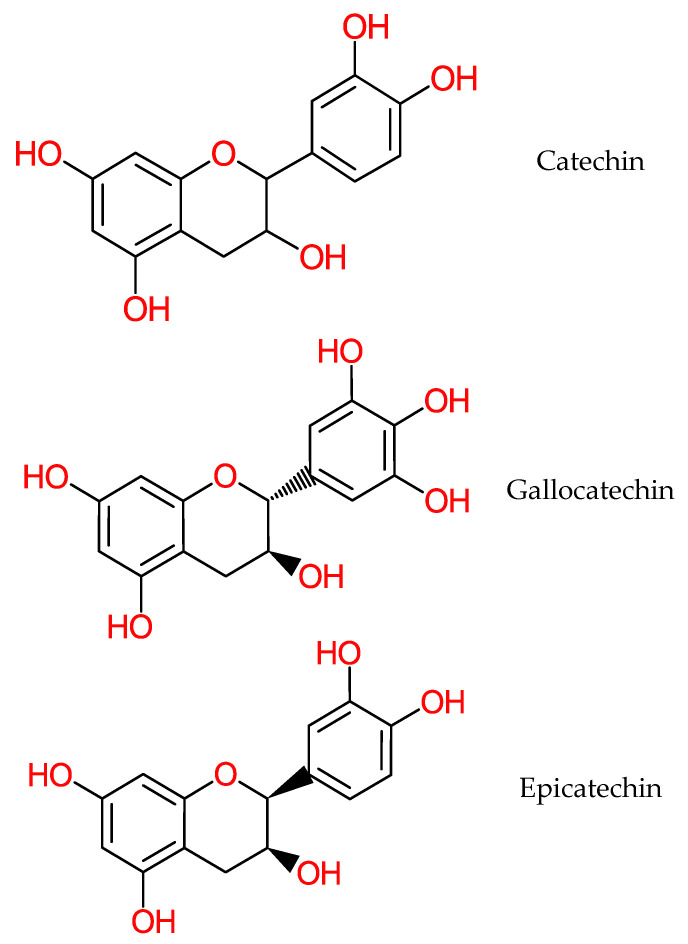
Structure of Catechin structures.

**Figure 17 molecules-25-05036-f017:**
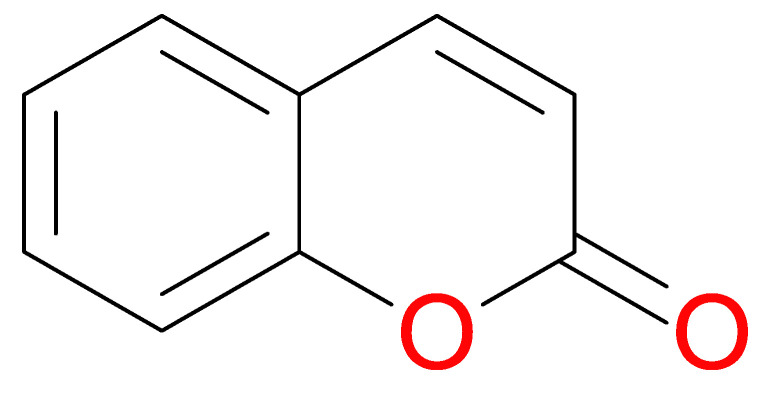
Coumarin structure.

**Figure 18 molecules-25-05036-f018:**
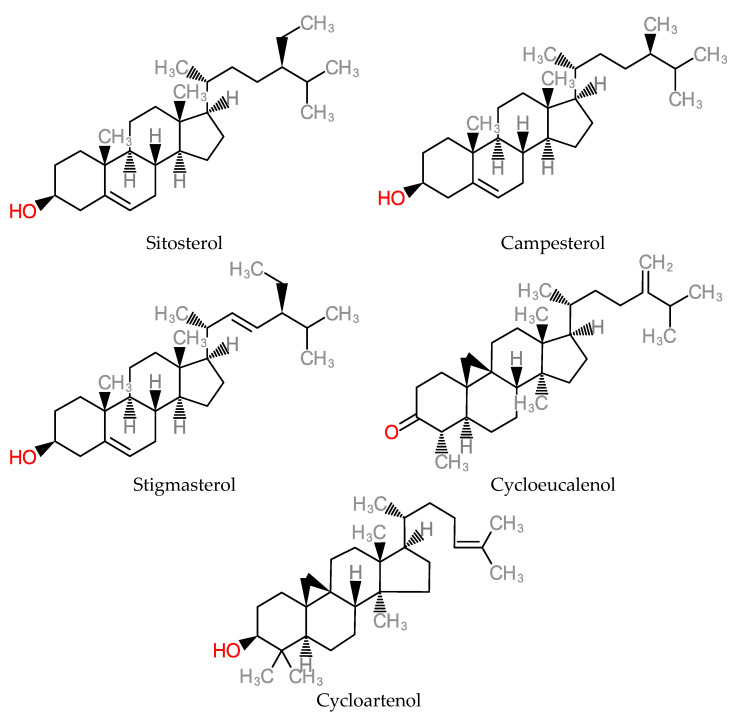
Structure of Phytosterols.

**Figure 19 molecules-25-05036-f019:**
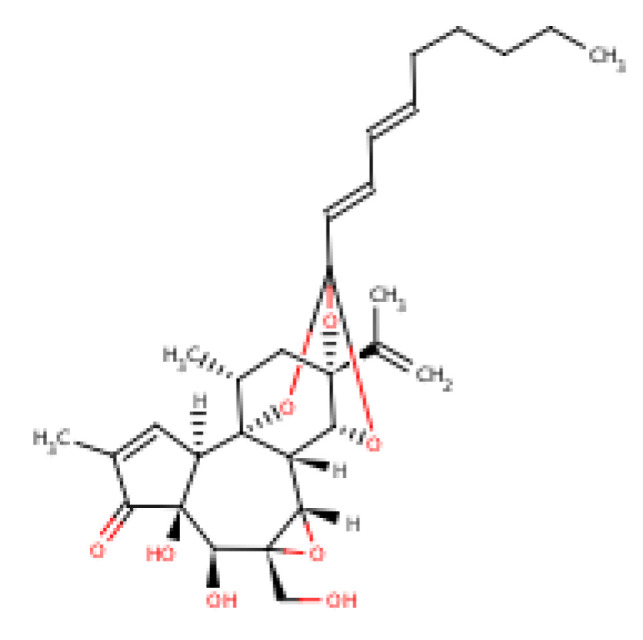
Terpenoids.

**Figure 20 molecules-25-05036-f020:**
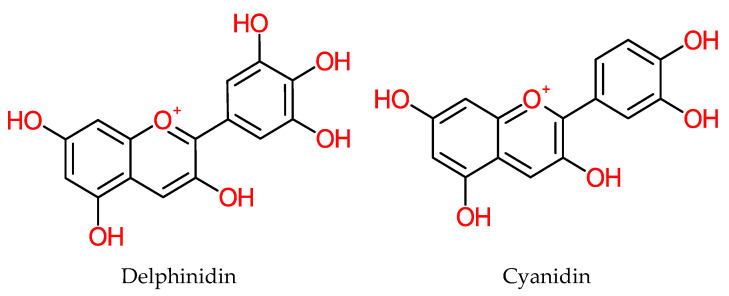
Structure of Anthocyanins.

**Figure 21 molecules-25-05036-f021:**
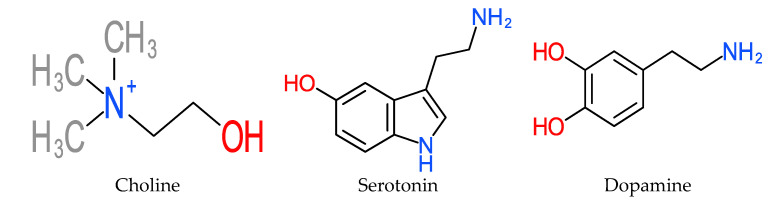
Structures of Catecholamines.

**Figure 22 molecules-25-05036-f022:**
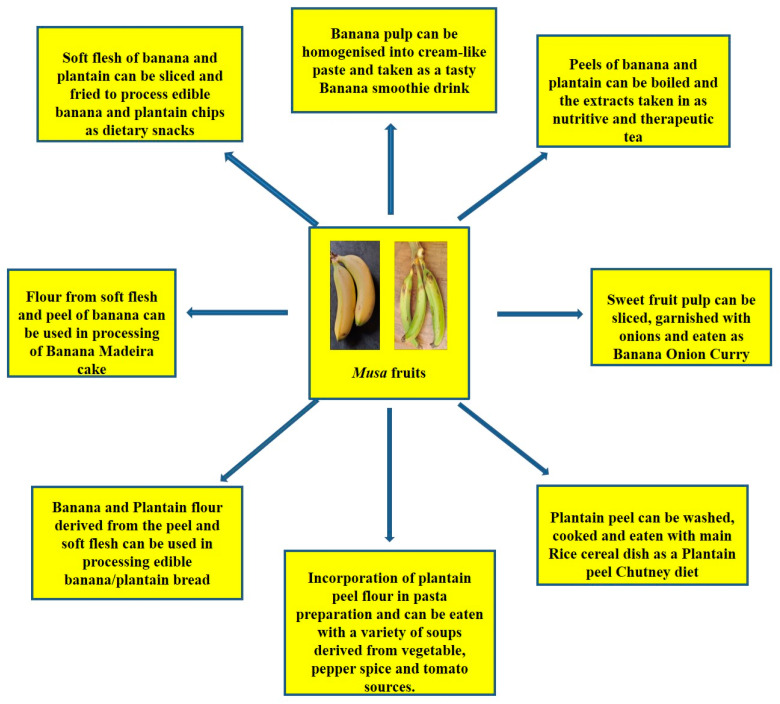
Dietary incorporation of *Musa* fruit compartments (soft flesh and peel) as contributors to improved diet diversity strategy towards alleviation to micronutrient deficiencies.

**Figure 23 molecules-25-05036-f023:**
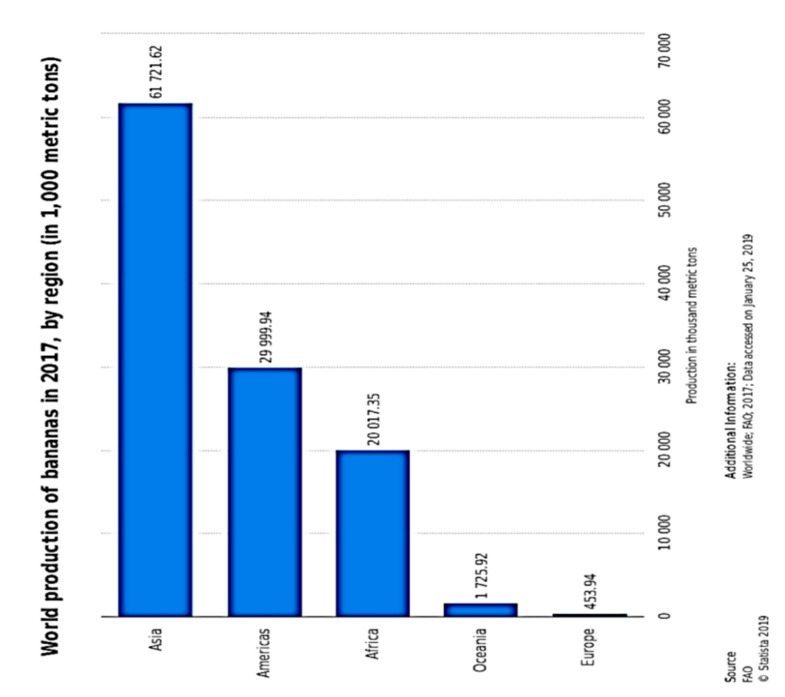
Global production of banana showing dominant banana output in the Asian, American and African continents [212].

**Table 1 molecules-25-05036-t001:** Comparative nutritional profile of fruit peels of *Musa* with other fruits.

Nutritional Factors	Other Fruit Peels (g/100 g)
Protein	> Pineapple, Mango, Orange, Apple, Pomegranate.
Carbohydrate	> Pawpaw, Watermelon
Ash content	> Pawpaw, Pineapple, Mango, Apple, Orange, Pomegranate, Watermelon
Calcium	> Pawpaw, Pineapple, Apple, Watermelon
Iron	> Mango, Pomegranate
Zinc	> Mango, Apple, Pomegranate
Manganese	> Pawpaw, Pineapple, Apple, Orange, Pomegranate, Watermelon.

Source: [40].

**Table 2 molecules-25-05036-t002:** Micronutritional factors identified in fruit compartments of Musa species fruits including the analytical methods.

Micronutrients	Soft flesh (Pulp)	Peel
**Manganese**	*Musa* spp. [46] [*AAS- Atomic Absorption Spectrophotometry*]	*M. paradisiaca* [47] [*AAS-Atomic Absorption Spectrophotometry*]
**Zinc**	*Musa* spp. [48] [*C_18_RP-HPLC; Microtitre Plate Spectrophotometry; Inductively Coupled Plasma- Optical Emission Spectrometry ICP-OES*]	
**Iron**	*Musa* spp. [48] [*C18RP-HPLC; MicrotitrePlate Spectrophotometry; ICP-OES*]	*Musa* spp. [48] [*C_18_RP-HPLC*; *Microtitre Plate Spectrophotometry;ICP-OES*]
**Copper**	*M. paradisiaca* [49]	
**Boron**	*Musa* (3.72 mg/kg) [50] [*Neutron γ-ray activation analysis*]	
**Phosphorus**	*M. sinensis* and *M. paradisiaca* [39] [*ICP-OES*]	*M. sinensis* and *M. paradisiaca* [39,51] [*ICP-OES*]
**Thiamine, Riboflavin, Niacin, Folate, Pantothenic acid and Pyridoxine**	Banana and Plantain (*Musa* spp.) [32]	

**Table 3 molecules-25-05036-t003:** Dietary phytoconstituents detected in the fruit compartments of *Musa* species and the analytical methods.

Dietary Phytoconstituents	Fruit Compartments		Protein/Gene Targets Linked with the Bioactive Dietary Compounds
	**Soft flesh (Pulp)**	**Peel**	
**Carotenoids**	-Provitamin A carotenoids [α-carotene (104.9 µg/100 g) and β-carotene (96.9 µg/100 g)] detected in orange coloured banana (*Musa* sp.) [117]. [*HPLC*] -α-carotene (61–1055 µg/100 g), trans β-carotene (50–1412 µg/100 g) and cis β-carotene (6–85 µg/100 g) detected across 18 cultivars of banana and plantain [118]. -Carotenoid content (0.130–0.159 mg/100 g) across developmental stages of edible banana [119]. [*Ultraviolet Spectrophotometry*]	-Lutein equivalent carotenoid content (3–4 µg/g), as well as other carotenoid components such as α-carotene, β-carotene, neoxanthin, α- cryptoxanthin and β-cryptoxanthin the peel of banana [120]. [*HPLC*]	
**Tocopherols**	-Total tocopherol (α-tocopherol, β-tocopherol, γ-tocopherol and δ-tocopherol) content (0.15 ± 0.09 mg/100 g) detected in banana [121]. [*NPLC-Normal Phase Liquid Chromatography*]	-β-tocopherol and Vitamin E [122]. [*GC-MS Gas Chromatography Mass Spectrometry*]	
**Catechins**	-Catechin in ripe and unripe banana cultivars [33]. [*HPLC*-*High Performance Liquid Chromatography*] [123] [*Ultraviolet visible Spectrophotometry and Liquid Chromatography* (*LC*)]. -Gallocatechin in banana soft flesh [124]. [*Thin-Layer Chromatography* (*TLC*) *and NMR*-*Nuclear Magnetic Resonance*].	-Catechin detected in the peels of ripe and unripe banana cultivars [33,123]. [*HPLC*- *High Performance Liquid Chromatography*]. -Catechin (30.21 mg/100 g) content in banana peel [125]. [*HPLC*-*High Performance Liquid Chromatography*]. -Gallocatechin (160 mg/100 g dry weight) in banana peel [126]. [*HPLC*-*HighPerformance Liquid Chromatography*]. -Epicatechin and gallocatechin detected in banana peel [122]. [*GCMS-Gas Chromatography-Mass Spectrometry*]. -Epicatechin in banana (*Musa* sp.) peel flour (1.11 ± 0.10 µg/g–4.13 ± 0.83 µg/g dry weight) across its Luvhele, Mabonde, M-red and Williams cultivars [127]. [*LC-MS-ESI Liquid Chromatography Electrospray Ionization*]	-Activator Protein-1 (AP-1) [128]. -Cyclooxygenase-2 (COX-2) [129]. -Caspases-3 [128]. -Caspases-10 [128]. -Fas [128]. -NF-κBp 105 [128].
**Coumarins**		-3-carboxycoumarin (0.79 mg/100 g) in banana peel [125]. [*HPLC-High Performance Liquid Chromatograpy*].	
**Phytosterols**	-Total Sterols (471 ± 38 mg/kg dry weight) in *M. paradisiaca* [130]. [*GC-MS Gas Chromatography Mass Spectrometry*]. -Phytosterols (2.8–12.4 g/kg dry weight) in unripe banana [131]. -Phytosterol members such as cycloeucalenol, cycloartenol, cyclo-eucalenone, stigmasterol, campesterol and β-sitosterol detected (2.8–12.4 g/kg dry weight) across unripe cultivars of *M. balbisiana* and *M. acuminata* [132]. [*GC-MS Gas Chromatography Mass Spectrometry*]	-β-sitosterol constituent in banana [133]. [*TLC-Thin Layer Chromatography* and *GLC-Gas Layer Chromatography*].	
**Terpenoids**		-Terpenoid content in banana (*M. sapientum*) and plantain (*M. paradisiaca*) [134]. -Terpenoid content detected in *M. paradisiaca* peel on the basis of three solvent extracts (aqueous, ethanol and chloroform) [135]. [*TLC-Thin Layer Chromatography*].	- Tumor proteins [136].
**Anthocyanins**	-Anthocyanin content in the soft flesh (pulp) (0.02 µg/g–0.16 µg/g fresh weight) of the red Hongjiaowang to yellow Baxijiao banana cultivars [137]. [*UPLC-PDA-QT_O_F-MS* and HPLC-PDA].	-High anthocyanin content in the peel (23.75 µg/g–154.75 µg/g fresh weight) of the red Hongjiaowang to yellow peel of Baxijiao banana cultivars [137]. [*UPLC-PDA-QT_O_F-MS* and HPLC-PDA] -Delphinidin and cyanidin in banana [138]. -Anthocyanin detected in *M. acuminata* peels [139]. [*pH Differential Simple Spectrophotometry*].	
**Catecholamines**	-Choline [131]. -[140]. -Catecholamine in pulp of *M. acuminata* and *M. paradisiaca* [141]. -Norepinephrine and serotonin in *M. paradisiaca* [142]. [*Spectrofluorophotometry*]. -Dopamine derivative of catecholamines in pulp of *M. cavendishii* (2.5–10 mg per 100 g) [143].	-High catecholamine content in extracts of banana peel [139]. [*LC- Liquid Chromatography*]. -Catecholamine content in ripe banana (*Musa* sp.) peels [143]. -Dopamine derivative of catecholamines in peel of *M. cavendishii* (80–560 mg per 100 g) [143].	-β-(β3)-receptors [144].

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
