# Peer review of "Potentials of Musa Species Fruits against Oxidative Stress-Induced and Diet-Linked Chronic Diseases: In Vitro and In Vivo Implications of Micronutritional Factors and Dietary Secondary Metabolite Compounds"

_molecules, 2020, doi:10.3390/molecules25215036_

Round 1

Reviewer 1 Report

   Oyeyinka and group have reviewed the biological activity of Musa fruits in the body, which is a correlation with their constituent micronutrient factors and dietary polyphenolic constituents such as minerals, vitamin members, anthocyanins, lutein, α-,β- carotenes, neoxanthins and cryptoxanthins, epi- and gallo catechins, catecholamines, 3-carboxycoumarin, β-sitosterol, monoterpenoids. As banana and plantain are very popular, reviewing its biological activities are important. I suggest acceptance after revision   1. Introduction: Expand with summarizing the statistics on the consumption of fruits and chemoprevention etc. Also, Epidemiological studies related to the consumption of fruits, micronutrients, and the risk of diseases. 2. Can you include pictures of all fruits that are consumed in one page? It will be very effective visually.   3. References: Check the numbering, they are numbered two times.  4. Chemical structures: You cannot copy structures from different sources. Draw structures in some chemical software and then paste it in the manuscript so that they look similar. Many of them available freely.  5. In many instances, the authors wrote that a particular compound has a particular activity. That is not enough. The authors need to write the mechanism proposed in terms of proteins etc, animal models or cell lines used, IC50 values.   6. Please add a column to your existing table 4 summarizing the protein or gene targets 

Reviewer 2 Report

Opinion related to the paper entitled: “The role of catechins in cellular responses to oxidative stress”.

The paper is written very carelessly (figures 2, 14 and many others). Figure 16 was copied from another work. There is no information if Authors have obtained permission from the journal (Molecules) or from the authors to use the fragment of their work. Excessive use of capital letter also where it is not needed. I don’t understand the meaning of tables 3 and 4.

I think that in this version, the work absolutely cannot be published in Molecules. The scientific standard of this paper is to low. Authors should see other review papers, write new paper, much more carefully. I can’t recommend publishing this work.

Reviewer 3 Report

In this manuscript, Oyeyinka and Afolayan have analyzed the potentials of Musa species fruits against oxidative stress-induced and diet-linked chronic diseases. Overall the topic is interesting and the review is comprehensive, informative and potentially of interest to a wide range of audiences. In the work there are some sentences that should be clarified and some general issues that should be addressed by the Authors during the revision of the manuscript. The changes and suggestions are listed below:

  1. Please do not report only the action of the analyzed class of molecules or compound present in the Musa species, but supply the concentration useful to reach these results
  2. As far as antioxidant activity are concerned, please reported the IC50
  3. Increase the quality of figure 23
  4. In figure 22 some word are cut
  5. The quality of the chemical structure is very low, please revise with an appropriate program to draw the chemical structure
  6. Revise overall the manuscript for the presence of several typing and grammar errors.

Round 2

Reviewer 1 Report

Authors have responded to my comments and added specific information is included as needed. The manuscript can be accepted.